# Content-Aware Few-Shot Meta-Learning for Cold-Start Recommendation on Portable Sensing Devices

**DOI:** 10.3390/s24175510

**Published:** 2024-08-26

**Authors:** Xiaomin Lv, Kai Fang, Tongcun Liu

**Affiliations:** 1School of Information Technology, The Zhejiang Shuren University, Hangzhou 310015, China; 2School of Mathematics and Computer Science, The Zhejiang A&F University, Hangzhou 311300, China; kaifang@zafu.edu.cn

**Keywords:** recommender system, cold start, meta-learning, representation learning

## Abstract

The cold-start problem in sequence recommendations presents a critical and challenging issue for portable sensing devices. Existing content-aware approaches often struggle to effectively distinguish the relative importance of content features and typically lack generalizability when processing new data. To address these limitations, we propose a content-aware few-shot meta-learning (CFSM) model to enhance the accuracy of cold-start sequence recommendations. Our model incorporates a double-tower network (DT-Net) that learns user and item representations through a meta-encoder and a mutual attention encoder, effectively mitigating the impact of noisy data on auxiliary information. By framing the cold-start problem as few-shot meta-learning, we employ a model-agnostic meta-optimization strategy to train the model across a variety of tasks during the meta-learning phase. Extensive experiments conducted on three real-world datasets—ShortVideos, MovieLens, and Book-Crossing—demonstrate the superiority of our model in cold-start recommendation scenarios. Compared to MetaCs-DNN, the second-best approach, CFSM, achieves improvements of 1.55%, 1.34%, and 2.42% under the AUC metric on the three datasets, respectively.

## 1. Introduction

The proliferation of portable sensing devices, such as smartphones and tablets, has made the acquisition of multimedia content an integral part of everyday life. However, the vast amount of available information on these devices complicates the process of identifying relevant items, often leading to information overload. Enhancing the user experience in this context requires leveraging behavioral data collected by sensors to model user interests and deliver targeted recommendations. Over the recent few decades, significant research has focused on developing recommender systems [1,2] that strive to make accurate predictions by modeling user preferences and item characteristics based on historical interaction data, such as ratings and clicks [1,3]. To more effectively capture users’ behavioral patterns, the task of sequential recommendation has emerged and attracted considerable research interest. These approaches aim to model user preferences by analyzing their historical interaction behaviors, enabling the prediction of subsequent items with which users are likely to engage [2,4,5,6]. Previous methods have utilized a variety of techniques, including Markov chains, recurrent neural networks, attention mechanisms, and graph neural networks, to model complex item transition patterns, demonstrating promising results. However, the performance of these methods often diminishes when encountering new items or new users who lack sufficient historical data—a challenge commonly referred to as the “cold-start” problem. This issue is particularly pertinent as new users continuously join online recommender systems, and new items are regularly introduced into their respective databases. Addressing the cold-start problem remains a critical challenge in the advancement of recommender systems. The availability of large amounts of item information data (e.g., publicly available reviews, descriptions, and images) provides potential solutions to the cold-start problem, and several content-aware approaches have been proposed to model latent relationships between users and items [7]. For example, Zhao et al. [8] leveraged knowledge extracted from social networking sites for cross-site cold-start recommendations; Liao et al. [9] extracted semantic representations for various deep multimodal rank-learning methods to improve the accuracy and robustness of recommender systems. However, the major limitation of these approaches is that the learned models may recommend the same items to users with similar content, thereby neglecting the personal interests implied in their sequential behaviors.

Another approach to mitigating the cold-start problem has been developed from the perspective of model architecture, eliminating the need for additional external information. For instance, Volkovs et al. [10] proposed a deep neural network (DNN) that generalizes missing information by employing dropout operations. However, this method struggles with robustness when interaction data are extremely sparse. Inspired by recent advancements in few-shot learning, meta-learning [11] has emerged as a promising technique for training models that can generalize to new tasks with only a few labeled training examples. This makes it particularly well suited for cold-start recommendation tasks, where only a limited amount of a user’s historical interaction data are available for training [12,13,14]. For example, the model proposed in [15] employs meta-learning to address the cold-start problem associated with the continuous arrival of new items, leading to enhanced performance in Twitter recommendations. Similarly, the models in [16,17] utilize meta-learning to predict the ratings of new items by learning user and item embeddings from user profiles and item descriptions. The model presented in [18] extracts user preferences from limited interactions and learns to match the targeted cold-start items with potential users. While these approaches alleviate the cold-start problem to some extent, they are prone to introducing noisy data by failing to effectively differentiate the importance of features. This can lead to suboptimal outcomes and ultimately limit the accuracy of recommendations.

In this paper, we aim to enhance cold-start recommendation accuracy by addressing the challenge of eliminating the influence of noisy data. We introduce a content-aware few-shot meta-learning (CFSM) model that incorporates a double-tower network (DT-Net) for learning user and item representations through a meta-encoder and a mutual attention encoder. The DT-Net is designed with two key components: the meta-encoder and the mutual attention encoder. The meta-encoder is responsible for learning users’ inherent interest representations based on their attribute features, while the mutual attention encoder captures item content representations from heterogeneous content information, effectively reducing the impact of noisy features through attention mechanisms. To train the CFSM model, we adopt a model-agnostic meta-optimization strategy during the training phase. This approach allows our model to generate representations for new users based on their attribute features, even in the absence of interaction data. As the model accumulates a small amount of interaction data, it is further refined through a few learning steps, enabling fast online recommendation services. The major contributions of this work are as follows:We design a DT-Net learning system that minimizes the introduction of irrelevant features, which could otherwise degrade model performance.We implement a model-agnostic meta-optimization strategy that empowers the CFSM to represent new users and items, thereby addressing the cold-start problem from both the model and auxiliary data perspectives.Empirical results on the three datasets demonstrate that CFSM outperforms state-of-the-art models in terms of the AUC metric.

The remainder of this paper is organized as follows: Section 2 reviews related works; Section 3 explains the preliminaries and definition; Section 4 describes the designed CFSM model; Section 5 details the experimental configuration; Section 6 presents the experimental results and analyses; and Section 7 provides the conclusion.

## 2. Related Works

To combat the cold-start issue, content data are commonly employed to infer the latent connections between users and items, significantly enhancing recommendation quality [19,20]. In a prior study [21], content attributes from LDA were merged into PMF. Subsequently, word2vec models were employed to tackle the cold-start problem as well [22]. Nevertheless, these content-centric strategies suffer from shortcomings due to their loose feature integration, leading to restricted enhancements in recommendations. Acknowledging the potential of deep learning to refine recommendation systems, Wang et al. [23] introduced a collaborative deep learning model for rating predictions. Kim et al. [24] devised a convolutional matrix factorization (MF) model, while Zhang et al. [25] seamlessly amalgamated the learning of latent user–item relationships with semantic representations from a knowledge base.

Recent progressions have confronted the cold-start dilemma through cross-domain recommenders that leverage pertinent source-domain data to alleviate sparsity in target-domain data. For instance, a neighborhood-based cross-domain latent feature mapping technique in [26] learns a customized feature mapping function for each cold-start user, along with a cross-modal contrastive learning framework for user cold-start recommendations [27]. Furthermore, Zhang et al. [25] proposed linking users between social media and e-commerce platforms, transforming their social networking characteristics into alternative representations to enhance product recommendations. Another tactic for the cold-start issue involves a transfer network approach [28]. For instance, Tay et al. [29] introduced a multi-pointer co-attention network leveraging an attention mechanism to address the new-item cold-start problem. However, cross-domain recommender systems often lack accuracy and may lead to improper suggestions.

Inspired by recent strides in few-shot learning, meta-learning has been integrated into recommender systems to tackle the cold-start hurdle. This strategy centers on learning a global parameter as a starting point for recommender model parameters. A personalized parameter is then locally adjusted to capture individual user preferences, while the global parameter is refined by minimizing the overall loss across diverse user-specific training tasks [11]. For instance, Vartak et al. [15] proposed two network architectures applying meta-learning tactics to forecast user ratings for Twitter posts based on past item rating data. Li et al. [30] framed the cold-start recommender as a zero-shot learning task. Lee et al. [16] fashioned a model to pinpoint trustworthy empirical candidates for custom preference estimations. Dong et al. [17] devised memory matrices to evade local optima for users exhibiting specific patterns. Wang et al. [31] introduced a preference learning decoupling framework empowered with meta-augmentation for user cold-start recommendations.

## 3. Preliminaries and Definition

***Preliminaries:*** For ease of representation, we first introduce some key mathematical notations. Following conventional symbol notation, we use uppercase bold letters to denote matrices (e.g., E), lowercase bold letters to denote row vectors (e.g., p), and regular typeface letters to represent scalars. We use calligraphic letters to represent sets (e.g., user set U) and |•| to denote the size of the set. Table 1 lists the major notations used in this paper.

We denote $U=\left\{u_{1},u_{2},\ldots ,u_{k},\ldots ,u_{\left|U\right|}\right\}$ as the set of users, $V=\left\{v_{1},v_{2},\ldots ,v,\ldots ,v_{\left|V\right|}\right\}$ as the set of items, $H=\{H_{1},H_{2},\ldots ,H_{\left|U\right|}\}$ as the interaction sequence set for all users, and $H_{i}=\left\{v_{i,1},v_{i,2},\ldots ,v_{i,k}\right\}$ as the chronologically ordered set of items that interacted based on the timestamps of user $u_{i}$. For cold-start scenarios, the user only interacts with a small number of items, and only a few users interact with each item. Thus, the length of the interaction sequence is very short.

***Objective:*** The objective is to enable the model to predict whether user $u_{i}$ will interact with item $v_{j}$ at a future time $t_{k+1}$ under the condition of data scarcity, which is a classical click-through rate (CTR) prediction task, and written as:(1)$${\hat{y}}_{ij}=P\left(r_{ij}=1|p_{i},q_{j}\right)=F_{\theta }\left(p_{i},q_{j}\right)\;\forall i\in U,\;j\in V$$
where $p_{i}$ and $q_{j}$ are the interest representation of user $u_{i}$ and the content representation of item $v_{j}$, respectively. $F_{\theta }\left(•\right)$ is the designed representation learning network for the recommender.

## 4. Proposed Approach

Owing to the scarcity of interaction data with new users and items, we regard the cold-start problem as a few-shot learning problem. Figure 1 illustrates the overall architecture. Figure 1a describes the meta-learning framework of the CFSM model, which aims to train the model with few-shot interaction data, and Figure 1b illustrates DT-Net, which aims to learn the representations of users and items from heterogeneous content information. In the following subsections, we explain the designed DT-Net and the meta-learning framework.

### 4.1. DT-Net for Representation Learning

DT-Net consists of a meta-encoder and a mutual attention encoder, as illustrated in Figure 1b. The meta-encoder learns the user’s inherent interest representation from their attribute features (e.g., age, gender, and location), and the mutual attention encoder learns the item’s content representation from heterogeneous content information (e.g., reviews, descriptions, tags, and pictures).

#### 4.1.1. Meta-Encoder

Extant studies [32] have demonstrated that projecting a classifiable variable into a dense latent semantic space can significantly improve the expressiveness of features. Thus, we first project the user’s attribute information into a dense semantic space using the embedding index technique. Specifically, the attribute set of the *i*-th user is denoted as $f_{i}=\left\{f_{i,1},f_{i,2},\ldots ,f_{i,k},\ldots ,f_{i,n}\right\}$, where *n* is the total number of attributes. Assuming that there are $n_{k}$ candidate values for the *k*-th attribute, we project each value into a *d*-dimensional semantic vector using the embedding index technique from the embedding matrix $E^{k}\in R^{n_{k}\times d}$. Subsequently, we generate the user’s feature vector by concatenating all semantic vectors:(2)$$u_{i}=\left[E{\left[f_{i,1}\right]}_{1};E{\left[f_{i,2}\right]}_{2};\ldots ;E{\left[f_{i,n}\right]}_{n}\right],$$
where $u_{i}\in R^{nd}$ and $E^{k}$ are learnable parameters. Subsequently, the user’s inherent preference representation is obtained using an *M*-layer feedforward fully connected neural network. The output of the *k*-th neural network layer is calculated as
(3)$$h_{k}=Swish\left(W_{k}^{T}h_{k-1}+b_{k}\right),$$
where k≥1, $h_{0}=u_{i}$, and Swish denotes the activation function. Compared with a rectified linear unit, Swish avoids the problem of gradient disappearance in deep neural networks (DNNs). Finally, we regard the output of the last layer, $h_{M}$, as the user’s inherent preference representation, $p_{i}=h_{M}$.

#### 4.1.2. Mutual Attention Encoder

Previous works [9,33,34] utilized auxiliary information (e.g., reviews, tags, descriptions, and images of the items) to enhance the representability of cold-start items by directly incorporating their content features into the recommendation model. However, it is difficult to distinguish important features from vast noise data when only a few pieces of information in the text and image can be discerned. Moreover, the introduction of irrelevant features can undermine the model’s ability to learn new items. To overcome this, we focus on important textual and image information by designing a new mutual attention encoder.

Specifically, we combine reviews, tags, and descriptions of the *j*-th item into a pseudo-document that we assume contains *L* words. We first project each word into a *w*-dimensional semantic vector using the embedding index technique. Typically, word embeddings are pretrained with narrative corpora or initialized randomly and optimized during training. We trained our model with the Google News corpus using word2vec, which showed the best performance in preliminary tests. The embedding matrix, $D_{j}\in R^{L\times w}$, for the pseudo-document is generated by concatenating the word embeddings in the order of their appearance. To learn the document features, a bidirectional gated recurrent unit (GRU) is used to capture the contextual information of words simultaneously. For each time step *t*, the output of the GRU is the concatenation of ${\vec{h}}_{t}$ and $h_{t}$, denoted as $h_{t}$, followed by a linear projection. The document feature matrix $H\in R^{L\times d}$ is generated by concatenating all hidden layers. For the image of the *j*-th item, following the work [9], we resize it to 224×224×3 and adopt VGG16 to extract the visual features. Note that only the last convolution layer is used. The visual feature matrix, $M\in R^{R\times d}$, for the item is generated with a single-layer full connection network, and *R* is the number of feature regions.

To distinguish the importance of each word in the pseudo document and the feature region in the image, a mutual attention mechanism is designed. The input of the attention layer consists of a query, key, and value, and the output of the attention network is the weighted sum over the value for which the weight matrix is determined by the queries and their corresponding keys. In our case, the attention layer takes the document feature matrix H and the visual feature matrix M as inputs, and we project them into the query and key via a nonlinear transformation with shared parameters. The correlation matrix is then calculated with a scaled dot product as
(4)$$A=\mathrm{softmax}\left(\frac{QK^{T}}{\sqrt{d}}\right),$$
(5)$$Q=\tan h(HW^{Q}+B_{q}),$$
(6)$$K=\tan h(MW^{Q}+B_{q}),$$
where $W^{Q}\in R^{d\times d}$, $W^{K}\in R^{d\times d}$, $B_{q}\in R^{1\times d}$, and $B_{k}\in R^{1\times d}$ are shared model parameters, and $\sqrt{d}$ is used to scale the dot product. The output of Equation (Equation 4) is an L×R normalized correlation matrix that represents the correlation between *L* words and *R* regions. By conducting max-pooling operations on rows and columns, we retain only the highly relevant and influential attention weights from the image and text, which are denoted as
(7)$$a_{H}=\;\max_{row}\left(A\right),a_{M}=\;\max_{col}\left(A\right),$$
where $a_{H}\in R^{L\times 1}$ and $a_{M}\in R^{R\times 1}$. Afterward, the output of the mutual attention encoder is defined as follows:(8)$$I=[{Q^{T}a}_{H};K^{T}a_{M}],$$
where $I\in R^{2d\times 1}$ is the latent content feature vector of the item, and the inherent attribute representation $q_{j}$ of the *j*-th item is obtained through the multilayer nonlinear mapping function of Equation (Equation 3).

Finally, the element-wise dot product of $p_{i}$ and $q_{j}$, with a sigmoid function, is used to generate the recommendation.

### 4.2. Optimization with Meta-Learning

We endow our CFSM model with the ability to quickly adapt to new users and items with sparse interaction data. To accomplish this, MAML training is accomplished with a series of tasks as a few-shot meta-learning problem. For each user, we define task T as the learning of the user’s interest representation and the item’s attribution representation from their interaction history data through our DT-Net. Let P(T) denote the probability distribution over all tasks and $T_{i}$ denote the task for user $u_{i}$. We generate support set $H_{i}^{s}=\left\{v_{i,1},\ldots ,v_{i,\frac{k}{2}}\right\}$ and query set $H_{i}^{q}=\left\{v_{i,(\frac{k}{2}+1)},\ldots ,v_{i,k}\right\}$ from the user’s interaction history, where both sets contain only a few samples.

Figure 1a illustrates the entire process. For each task $T_{i}$, the optimization process is divided into two phases: local gradient searching and global parameter updating. For the first stage, we randomly initialize parameter θ and calculate the internal loss $L_{T_{i}}^{s}$ on support set $H_{i}^{s}$. The updated parameter $\theta_{i}^{*}$ is computed using one gradient descent update on task $T_{i}$, calculated as
(9)$$\theta_{i}^{*}=\theta -\alpha \nabla_{\theta }L_{T_{i}}^{s}\left(F_{\theta }\left(p_{i},q_{j}\right)\right),$$
where $L_{T_{i}}^{s}\left(•\right)$ is the loss of task $T_{i}$. Cross-entropy loss is used in our method, and step size α may be fixed as a hyperparameter or meta-learned. For simplicity of notation, we only consider one gradient descent. This stage aims to optimize the model parameters so that one or a small number of gradient steps on a new task will produce maximally effective behaviors. The local gradient search stage can be seen as a process of exploring user preference with many user task sets, hoping that the model finds a potential optimization direction for learning how to generate the user’s preferred parameters. Finally, during the second stage, meta-optimization across tasks is performed via stochastic gradient descent, such that the model parameters θ are updated as follows:(10)$$\theta =\theta -\beta \nabla_{\theta }\sum_{T_{i}∼P\left(T\right)}L_{T_{i}}^{q}\left(F_{\theta_{i}^{*}}\left(p_{i},q_{j}\right)\right)$$
where β is the meta-step size. The complete algorithm is outlined in Algorithm 1.
**Algorithm 1** Model training of our proposed CFSM model.**Input:** P(T): distribution over tasks; α,β: step size hyperparameters  1:randomly initialize θ  2:randomly initialize the remaining parameter, δ  3:**while** not converge **do**  4:      Sample batch of task $T_{i}∼P\left(T\right)$  5:      **for all** $T_{i}$ **do**  6:            evaluate $\nabla_{\theta }L_{T_{i}}^{s}\left(F_{\theta }\left(p_{i},q_{j}\right)\right)$  7:            Compute fast gradients and inner update $\theta_{i}^{*}\leftarrow \theta -\alpha \nabla_{\theta }L_{T_{i}}^{s}\left(F_{\theta }\left(p_{i},q_{j}\right)\right)$  8:      **end for**  9:      Global update $\theta \leftarrow \theta -\beta \nabla_{\theta }L_{T_{i}}^{q}\left(F_{\theta_{i}^{*}}\left(p_{i},q_{j}\right)\right)$10:      Global update δ with gradient descent: $\delta \leftarrow \delta -\beta \nabla_{\delta }L_{T_{i}}^{q}\left(F_{\delta }\left(p_{i},q_{j}\right)\right)$11:**end while**

## 5. Experimental Settings

### 5.1. Datasets

Three datasets with different domains were used to demonstrate the efficacy of the proposed model. For all datasets, 15% of users were randomly selected as “new”. Table 2 presents the basic details of the datasets.

**ShortVideos** is a collection of Chinese short videos, and it contained three tables: interaction data, user information, and video information. The user information included age, gender, and location, and the video information included title, category, text description, and cover image. For each user, we retained K = 36 samples and sorted them by the timestamp. Users whose samples were fewer than K were removed.**MovieLens-1M** (https://grouplens.org/datasets/movielens/1m/) (accessed on 22 August 2024) is one of the most widely used benchmark datasets, and it included three tables: rating, user information, and movie information. It contains 1M rating records as-signed by users. The user information included ID, age, occupation, gender, and zip code. The movie information included title, publication, and genre, and we obtained text descriptions and cover images of the movies from the Internet Movie Database. We binarized the rating scores to zero and one, and a value greater than three was set to one.**Book-Crossing** (https://grouplens.org/datasets/book-crossing/) (accessed on 22 August 2024) is a public dataset containing three tables: rating, user information, and book information. User information included ID, age, and location, and book information included title, author, and publisher. We obtained text descriptions and cover images from Amazon.com. We binarized the rating score to zero and one, and a value larger than five was set to one.

### 5.2. Evaluation Settings

The detailed software and hardware environment of this experiment is introduced as follows. The experiments used PyCharm as the Python integrated development environment. All codes were written in Python 3.9 and relied on a variety of called packages. All deep learning models were built in the TensorFlow 2.6.0 framework. Additionally, the hardware environment included an i7-8750 H CPU and a GTX 1080 GPU. We used the Adam optimizer [35] for training during the meta-learning optimization step. The hyperparameters used in the model were α and β, which were tuned in [0.0005, 0.001, 0.02, 0.05], and the optimal values were 0.001 and 0.02, respectively. We also applied a dropout (0.5) to avoid overfitting. The meta-encoder featured four hidden layers for MovieLens-1M and ShortVideos, and eight hidden layers for Book-Crossing. A more elaborate parameter analysis will be provided in the subsequent sections.

To verify the performance of our CFSM model, we compared it to six traditional machine, deep, and meta-learning models, including the PMF model [36], a factorization machine (FM) [37], Deep FM [38], a dropout network (DropoutNet) [10], the two-tower DNN (TDNN) [39], and the meta-cold-start (metaCS) DNN [13]. All the compared models are briefly described as follows:PMF: This model obtains feature matrices by exploiting user–item interaction data and predicts unknown values in a rating matrix, which has been shown to be a particularly flexible and effective framework for addressing large, sparse, and very imbalanced datasets.FM: This model adopts linear and pairwise feature interactions as the inner products of their respective feature latent vectors. This approach is far more effective than previous approaches, especially with sparse datasets.DeepFM: This model combines the power of factorization machines for the deep learning of features in new recommender architectures. Low-order feature interactions are modeled, such as FMs and high-order feature interaction models (e.g., DNNs).DropoutNet: This model is based on the observation that cold starting is equivalent to the missing data problem in which preference information is missing. Instead of adding additional objective terms to the model content, this model modifies the learning procedure by applying a dropout to explicitly condition the model for missing input.TDNN: This model is the most widely used deep learning recommendation model in the industry, and it applies two parallel networks to learn latent vector representations of users and items.MetaCS-DNN: This model learns the representation of users and items using a DNN, and the parameters in the deep learning recommendation framework are updated by applying meta-learning optimization.

In this study, we implemented PMF and TDNN using their open-source codes and referred to the setup procedures provided in their corresponding papers. We trained all models under classical CTR-based cross entropy loss on all three datasets and fine-tuned their hyperparameters for their respective optimal performance. Therefore, all models could make recommendations simultaneously.

### 5.3. Metrics

We used the classic AUC metric to evaluate the performance of our CFSM against comparative methods [40,41]. The AUC reflects the probability that a CTR predictor will assign a higher score to a randomly chosen positive item than to a randomly chosen negative item. A higher AUC indicates better performance. Following a previous work [40], we adopted relative improvement (RelaImpr) to measure the relative improvements over the PMF baseline model. Note that all metrics were evaluated only for the selected test users.

## 6. Results and Discussion

### 6.1. Comparison of Cold-Start Scenarios

To comprehensively evaluate the performance of the proposed CFSM model, we were inspired by the work of [17] to design three experimental schemes: (1) cold users with existing items (C-W); (2) existing users with cold items (W-C); and (3) cold users with cold items (C-C). The latter is the most extreme scenario. The experimental AUC and RelaImpr results are shown in Figure 2 and Figure 3.

The results show that, under the three different experimental settings, our CFSM model significantly outperformed the comparative approaches on the MovieLens-1M, Book-Crossing, and ShortVideos datasets. Compared with the best baseline (MetaCS-DNN) for the C-W scenario, the performance improvements on the three datasets were 4.34%, 2.87%, and 5.68%, respectively. For the W-C scenario, the improvements were 1.32%, 1.26%, and 2.99%, respectively. For the C-C scenario, the improvements were 2.6%, 4.31%, and 5.17%, respectively. From the experimental results, we also found that the transitional machine learning recommendation methods (i.e., PMF and MF) performed the worst, but the deep learning approaches (i.e., DeepFM, TDNN, and DropoutNet) performed better. Although DeepFM and TDNN solved the data sparsity problem from the perspective of heterogeneous data, they required large-scale interactive data for model optimization and had difficulty dealing with the cold-start problem. DropoutNet utilizes a deep learning technique to fit the information lost caused by dropout regularization and uses it to supplement the lack of interactive information in a cold-start recommendation scenario. Moreover, we found that the methods based on meta-learning (i.e., MetaCS-DNN and MLTN) had obvious advantages in the cold-start scenarios. Compared with MetaCS-DNN, CFSM not only provided a gradient-based meta-optimization algorithm for parameter optimization but also accomplished heterogeneous data denoising with its mutual attention structure, which alleviated the problem of recommending new items, achieving the best experimental results under all three experimental schemes.

### 6.2. Comparison with the Normal Scenario

In addition to cold-start recommendations, we also examined the performance of our CFSM and the comparative approaches with traditional recommendations. The experimental results for the AUC metric and RelaImpr are listed in Table 3. The results showed that our CFSM model still achieved the best performance; however, the improvement was smaller than in the cold-start scenario. For example, compared with the best baseline (MetaCS-DNN), the AUC improvements for MovieLens-1M, Book-Crossing, and ShortVideos were 1.55%, 1.34%, and 2.42%, respectively.

### 6.3. Ablation Study

We conducted an ablation study to analyze the influence of meta-learning optimization components on the overall CSFM model’s performance. We compared the performance of CFSM with meta-learning optimization to a version of CFSM without meta-learning optimization (CSFM-naïve) on three datasets in the C-W scenario.

CSFM-naïve uses the same model structure as CSFM, but there is no parameter update process in the meta-optimization stage. The experimental results were calculated for the query set of the test user set. For CSFM-naïve, the query set of the test user set no longer underwent meta-learning optimization; however, the parameters of the trained model were fine-tuned. These results are shown in Table 4.

The performance of the CSFM model optimized by meta-learning was greatly improved in the cold-start scenario for new users. The initial parameters obtained by optimizing the second-order gradient in meta-learning can be understood as CSFM learning a set of prior knowledge; hence, it gained the ability to generalize well to new user recommendation tasks.

### 6.4. Parameter Analysis

In this section, we analyze the impact of the size of the support set, number of hidden layers, and dimension d.

Influence of the size of the support set: In our model, the support dataset was used for training; thus, the dataset’s size was strongly related to model efficiency. The sizes of the support datasets were set to 6, 12, 24, 36, 54, 66, and 80, and the results are shown in Figure 4a. With an increase in the support dataset size, the AUC metric increased slowly and flattened. The optimal size of the support set was obtained at 54 for MovieLens-1M and 36 each for Book-Crossing and ShortVideos.

Influence of the number of hidden layers: We further analyzed the influence of the number of hidden layers on the model’s performance. These numbers were set to 2, 4, 8, and 16, and the results are shown in Figure 4b. As the number of hidden layers increased, CFSM performance first increased; then, it decreased. However, it fluctuated only within a small range, indicating that the meta-learning optimization strategy is reliable and that no drastic fluctuations in performance occurred owing to changes in the number of hidden layers. A deeper network could enhance the ability of the CFSM model to capture user preferences by relying on a smaller dataset; however, increasing the number of hidden layers can also lead to overfitting and the vanishing gradient problem. Both negatively affect model performance. Thus, the best number was set to 4 for MovieLens-1M and ShortVideos and 8 for Book-Crossing.

Influence of the number of parameters: Figure 4c shows the impact of the number of parameters in the CFSM model on the three datasets. The number reflects the parameters of each layer in DT-Net. The dimensions of the first layer were tuned in [8, 16, 64, 256], and P denotes the total number of parameters under different dimensions. As shown in Figure 4c, as the number of parameters increased, the performance of the model first increased and then decreased.

## 7. Conclusions

Viewing the cold-start dilemma through the lens of a few-shot learning challenge, we introduce a novel cold-start recommendation methodology dubbed CFSM, integrating DT-Net and meta-learning. DT-Net is tailored to extract a user’s intrinsic preference cues from attribution features and an item’s content nuances from varied content sources. A mutual attention mechanism is deployed to counteract the influence of noisy data on auxiliary information. Additionally, a model-agnostic meta-optimization strategy is embraced to train the model across a spectrum of tasks in the meta-learning phase. Extensive experiments across three real-world datasets covering distinct cold-start recommendation scenarios showcased the consistent outperformance of our CFSM over comparative methods. Notably, it achieved superior performance even in traditional recommendation settings. The robustness of our CFSM model shines in its adeptness at handling data scarcity in cold-start recommendation scenarios, effectively mitigating issues like instability, slow convergence, and limited generalization. However, a notable limitation is the computational complexity linked to the meta-optimization approach, prompting a need for scalable implementations with regard to practical deployment. Our future initiatives will focus on honing a more resilient and efficient meta-optimization algorithm to enhance the effectiveness of our methodology in tackling cold-start recommendation challenges.

## Figures and Tables

**Figure 1 sensors-24-05510-f001:**
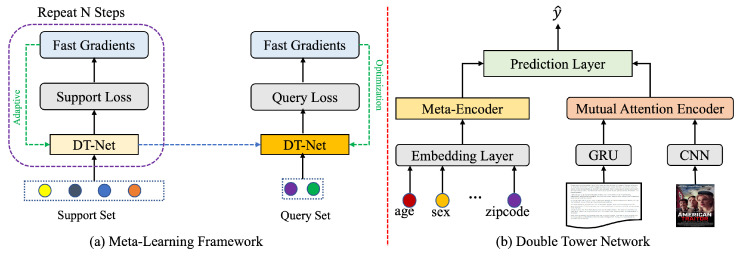
Overall workflow of the proposed CFSM model.

**Figure 2 sensors-24-05510-f002:**
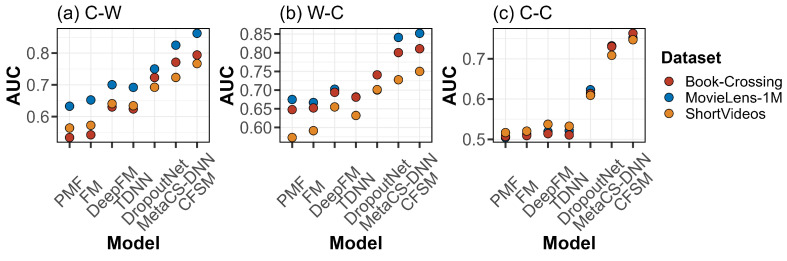
Experiment results on the three datasets using the area under the receiver operator characteristic curve (AUC) metric for different recommendation scenarios. Definitions: C-C, cold users with cold items; C-W, cold users with existing items; DNN, deep neural network; FM, factorization machine; MetaCS, meta-cold start; PMF, probabilistic matrix factorization; TDNN, two-tower DNN; W-C, existing users with cold items.

**Figure 3 sensors-24-05510-f003:**
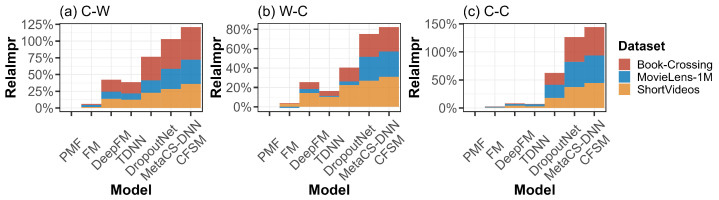
Experiment results on the three datasets using the relative improvement (RelaImpr) metric under different recommendation scenarios. Definitions: C-C, cold users with cold items; C-W, cold users with existing items; DNN, deep neural network; FM, factorization machine; MetaCS, meta-cold start; PMF, probabilistic matrix factorization; TDNN, two-tower DNN; W-C, existing users with cold items.

**Figure 4 sensors-24-05510-f004:**
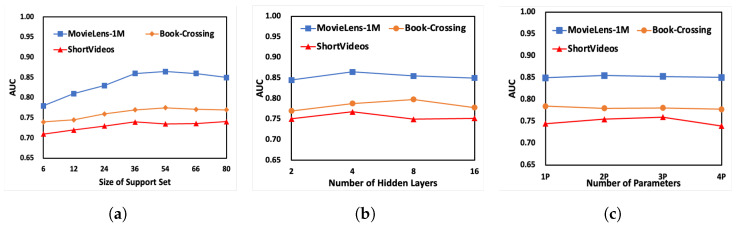
Influence of different parameters on model performance. (**a**) AUC score under different support set size. (**b**) AUC score under different numbers of hidden layers. (**c**) AUC score under different number of parameters.

**Table 1 sensors-24-05510-t001:** Key mathematical notations used in this paper.

Variable	Interpretation
U,V	Set of users and items
$H_{i}$	Sequence of user–item interaction
$p_{i}$	Interest representation of user $u_{i}$
$q_{j}$	Content representation of item $v_{j}$
$E^{k}$	Embedding matrix for user’s *k*th attribution
H,M	Document and visual feature matrices, respectively
Q,K	Attention network’s query and key, respectively

**Table 2 sensors-24-05510-t002:** Statistics of the datasets used in this paper.

Dataset	ShortVideos	MovieLens-1M	Book-Crossing
#Users	289,434	6040	278,858
#Items	19,412	3706	271,379
#Interactions	10,419,624	1,000,209	1,149,780
User information	ID, age, gender, location	ID, age, gender, occupation, zip-code	ID, age, location
Item information	title, category, description, cover image	title, category, description, posters	title, publisher, cover image, description

**Table 3 sensors-24-05510-t003:** Experimental results of our model vs. state-of-the-art recommenders.

	MovieLens-1M		Book-Crossing		ShortVideos	
	AUC	RelaImpr	AUC	RelaImpr	AUC	RelaImpr
PMF	0.7521	0.00%	0.7024	0.00%	0.6041	0.00%
FM	0.7400	−1.61%	0.7132	1.54%	0.6724	11.31%
DeepFM	0.8517	13.24%	0.7915	12.69%	0.7541	24.83%
TDNN	0.8247	9.65%	0.7821	11.35%	0.7142	18.23%
DropoutNet	0.7546	0.33%	0.7831	11.49%	0.7501	24.17%
MetaCS-DNN	0.8251	9.71%	0.8011	14.05%	0.7497	24.10%
**CFSM**	**0.8651**	**15.02**%	**0.8141**	**15.90**%	**0.7704**	**27.53**%
Δ%	1.55%		1.34%		2.42%	

**Table 4 sensors-24-05510-t004:** Results of ablation study on three datasets under the cold-users-with-existing-items scenario.

	MovieLens-1M		Book-Crossing		ShortVideos	
	AUC	RelaImpr	AUC	RelaImpr	AUC	RelaImpr
CSFM-naïve	0.7238	0.00%	0.6754	0.00%	0.6864	0.00%
CSFM	0.8651	19.52%	0.8141	20.54%	0.7704	12.24%
Δ%	14.13%		13.87%		8.4%	

## Data Availability

Data are contained within the article.

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
