# Peer review of "Content-Aware Few-Shot Meta-Learning for Cold-Start Recommendation on Portable Sensing Devices"

_sensors, 2024, doi:10.3390/s24175510_

Round 1

Reviewer 1 Report

Comments and Suggestions for Authors

4.1.2  what if it is greater than L words? For the image of the j-th item, why was the size of 224 × 224 × 3 adopted?

Need more details about Fig 1a. i.e. How Fast Adaptive works?

4.2 This step is repeated N times to find the proper direction. Please clarify "proper direction" and what vaule is for N? Also, confused about N and proper direction. N won't stop until a proper direction? Or a proper direction will be stated once reaching N?

For simplicity of notation, we only consider one gradient descent-----> Disagree with selections of one gradient decent--->simplifying notation.

5.1 The lack of sources of datasets does not convince readers. Can authors post datasets?

6.1 The latter is the most extreme scenario. Which one?

6.4 the results are shown in Fig. 2a? or 4a? Also examine the rest of indications of the same paragraph.

     However, it fluctuated only within a small range, indicating that the meta-learning optimization strategy is reliable..... Please quantify the relationship between reliable and fluctuation. Personally, I don't think it is a correct statement. In addition, this paper may not align adequately with the scope of Sensors journal, as it does not involve the use of sensors or related devices.

Author Response

Thank you very much for taking the time to review this manuscript. Please find the detailed responses below and the corresponding revisions in the resubmitted files.

Comments 1:  4.1.2  what if it is greater than L words? For the image of the j-th item, why was the size of 224 × 224 × 3 adopted?

Response 1: The "L" serves as a notation signifying the number of words post content merge, rather than a strict threshold. The dimensions of 224x224x3 are widely employed in existing literature; hence, we have opted for this standard size as well.

Comments 2: Need more details about Fig 1a. i.e. How Fast Adaptive works?

Response 2: We have removed the erroneous reference to "fast adaptive works" to prevent any potential misinterpretation.

Comments 3: 4.2 This step is repeated N times to find the proper direction. Please clarify "proper direction" and what vaule is for N? Also, confused about N and proper direction. N won't stop until a proper direction? Or a proper direction will be stated once reaching N?

Response 3: The sentence has been removed to prevent any misunderstanding.

Comments 4: 5.1 The lack of sources of datasets does not convince readers. Can authors post datasets?

Response 4:  Aside from the short video dataset, the remaining two datasets are publicly accessible.

Comments 5: 6.1 The latter is the most extreme scenario. Which one?

Response 5:  It is typo, the word is last

Comments 6: 6.4 the results are shown in Fig. 2a? or 4a? Also examine the rest of indications of the same paragraph.

Response 6:  We have rectified the error.

Reviewer 2 Report

Comments and Suggestions for Authors

I have reviewed your manuscript titled "Content-Aware Few-Shot Meta-Learning for Cold-Start Recommendation on Portable Sensing Devices" submitted to the "Sensors" journal. Your study presents an innovative approach to solving the cold-start problem in recommendation systems and makes significant contributions to the context of sensor networks and IoT applications. The originality and innovative value of the study are high. However, I believe some improvements are necessary.

Scope

The study proposes a content-aware few-shot meta-learning (CFSM) model to solve the cold-start problem and improve the accuracy of recommendation systems. The model uses a dual-tower network (DT-Net) to learn user and product representations and applies model-agnostic meta-optimization during the meta-learning phase. The authors conducted experiments on three real datasets, demonstrating that CFSM outperforms existing approaches in cold-start scenarios. Accordingly, the study appears to fit well within the scope of the "Advanced Mobile Edge Computing in 5G Networks" special issue of the Sensor Networks section of the Sensors journal.

Originality

The study highlights the cold-start problem, which arises when there is insufficient historical data for new users or items, negatively impacting the performance of recommendation algorithms. To address this issue, the authors employ a few-shot meta-learning approach. By combining a meta-encoder and a mutual attention encoder within a DT-Net structure, they effectively capture the features of the datasets to learn user and product representations. Compared to past research, this innovative solution to data scarcity offers significant contributions to the field.

Suggestions and Improvements

  1. References: The references are current, but they could be further updated with recent studies. An overview table summarizing past models related to the cold-start problem should be included, comparing these models with the proposed model.
  2. Datasets and Methodology: More detailed descriptions of the datasets used and model parameters should be provided. This will enhance the reproducibility of the study.
  3. Tables and Figures: Tables should be made more comprehensible, and the color contrasts and labels in the figures should be improved.
  4. Discussion of Results: The findings should be discussed in a broader context and compared with other studies in the literature. Practical applications of the findings should also be explained.
  5. Researchers' Previous Works: The authors have focused on cold-start systems and employed meta-learning and deep learning methods in their previous works. If this study continues their previous research, an overview table should highlight the differences and innovative aspects of this study.

Considering these feedback points and incorporating the suggested revisions will improve the quality and publishability of your manuscript. I wish you continued success in your research.

Sincerely.

Author Response

Thank you very much for taking the time to review this manuscript. Please find the detailed responses below and the corresponding revisions in the re-submitted files.

Comments 1:References: The references are current, but they could be further updated with recent studies. An overview table summarizing past models related to the cold-start problem should be included, comparing these models with the proposed model.

Response 1:  We have refreshed the latest studies. We believe an overview table for summarization is unnecessary, as this format is not utilized in any current work.

Comments 2:Datasets and Methodology: More detailed descriptions of the datasets used and model parameters should be provided. This will enhance the reproducibility of the study.

Response 2:   Appreciate your suggestion. We have enriched the dataset description in section 5.1 and integrated the dataset URL in the footnote for all datasets except for ShortVideos, sourced from the KuaiShou APP for a future release. The model parameter settings have been included in section 5.2.

Comments 3:Tables and Figures: Tables should be made more comprehensible, and the color contrasts and labels in the figures should be improved.

Response 3:  The Tables and Figures have improved.

Comments 4:Discussion of Results: The findings should be discussed in a broader context and compared with other studies in the literature. Practical applications of the findings should also be explained.

Response 4:  The comparison with previous studies was outlined in sections 6.1 and 6.2, while the application of the findings was included in section 7.

Comments 5: Researchers' Previous Works: The authors have focused on cold-start systems and employed meta-learning and deep learning methods in their previous works. If this study continues their previous research, an overview table should highlight the differences and innovative aspects of this study.

Response 5:  We have emphasized the distinctions from prior research and the innovative aspects of this study.

Round 2

Reviewer 1 Report

Comments and Suggestions for Authors

Good to be published.